# First Capture of a Jaguar Using a Minimally Invasive Capture System for GPS Tracking in an Isolated Patch of Atlantic Forest in Southern Brazil

**DOI:** 10.3390/ani13213314

**Published:** 2023-10-25

**Authors:** Francisco Palomares, Tarcízio Antônio Rego de Paula, Ana Carolina Srbek-Araujo

**Affiliations:** 1Doñana Biological Station, CSIC, Department of Conservation Biology, Avda. Américo Vespucio 26, Isla de la Cartuja, 41092 Sevilla, Spain; 2Departamento de Veterinária, Universidade Federal de Viçosa, Av. Peter Henry Rolfs, s/n, Campus Universitário, Viçosa 36570-900, Minas Gerais, Brazil; tarcizio@ufv.br; 3Programa de Pós-Graduação em Ecologia de Ecossistemas e Programa de Pós-Graduação emCiência Animal, Universidade Vila Velha, Av. Comissário José Dantas de Melo, 21, Boa Vista, Vila Velha 29102-920, Espírito Santo, Brazil

**Keywords:** habitat use, home range size, MICS, *Panthera onca*, remote capture, trapping technique

## Abstract

**Simple Summary:**

Safe and efficient capture of animals is a fundamental aim when working with large predators. Here, we describe a new system for capturing large Neotropical felids. For the first time, a minimally invasive capture system was used to capture a jaguar and facilitate its posterior GPS tracking in a fragment of Atlantic Forest in southeastern Brazil. The jaguar was a 16-year-old adult male, who mainly moved over an area of 175 km^2^ in protected areas. The GPS collar worked optimally, and it located the jaguar on 86% of expected occasions. Most locations were in native grassland, marsh, and dense lowland forest. Using a minimally invasive capture system is more efficient, selective, portable, and less risky for both animals and trappers, and causes less stress to animals compared to other capture methods used for the species.

**Abstract:**

This study presents the first successful capture using GPS tagging of a jaguar (*Panthera onca*) using a minimally invasive capture system (MICS). We used snare-foot traps and a MICS during two capture campaigns in a fragment of Atlantic Forest in southeastern Brazil. The specimen disarmed snares on different occasions, and capture was only possible with the MICS. The captured jaguar, an estimated 16-year-old adult male, was monitored using a GPS Vertex Plus Iridium collar with an optimal performance of 86% in expected locations. The jaguar’s home range (659 km^2^ by MPC and 174 km^2^ by 95%K) was within the observed range for the species and the animal was primarily maintained in protected areas. The habitat types most frequently used were native grassland (27.2% of 4798 fixes), marsh (24.8%), and dense lowland forest (24.7%). The use of a MICS for trapping jaguars is a promising technique that shows advantages in terms of efficiency, selectivity, portability, reduced potential risk of injury to animals or trappers, and animal stress compared to other capture methods used for the species.

## 1. Introduction

Large mammalian carnivores play a crucial role in maintaining biodiversity [1,2,3] but are facing significant threats from direct and indirect human activities [4,5]. The use of radio and GPS tracking techniques over the last few decades has revealed the impact of human activity on the spatial ecology of these species. However, capturing and immobilizing large carnivores for tracking is often challenging and unsafe [6,7,8,9,10], and traditional capture methods, including trained hounds, box-traps, and foot-snares, have their own disadvantages [11,12].

Palomares [10] proposed the use of a minimally invasive capture system (MICS [13]) to efficiently and safely capture large neotropical felids, and in this study, we report the first successful capture of a jaguar (*Panthera onca*) using this technique. With this method, we were able to safely capture one of the last jaguars in an isolated Atlantic Forest patch in Southeastern Brazil [14] with a relatively low investment of effort, time, and resources, while ensuring the safety of the animal and trappers. The MICS was developed and tested using a medium-sized felid, the European lynx (*Lynx lynx*), and has the potential to be an adequate capture method for large felids, such as jaguars and pumas. However, until now, no studies have been conducted to test this method for the capture of wild jaguars, pumas, or any other large mammal species, despite the a priori advantages over other, more conventional methods. The device consists of a blowgun remotely controlled by cameras, which is monitored and triggered from up to 400 m away. This method has been shown to be efficient, selective, and may be safe for animals and people [13].

We also describe the performance of GPS collars installed on a jaguar for the first time in this Atlantic Forest patch and the general use of the forest patch by the jaguar. Jaguars in the Atlantic Forest are critically threatened, requiring the development of safe methods for managing individuals. About 85% of the jaguar’s habitat in this biome has been lost and the species persists in low densities in less than 3% of the region [15].

## 2. Materials and Methods

### 2.1. Study Area

The capture was carried out in the Vale Natural Reserve (Reserva Natural Vale—RNV; 22,711 ha), located between the municipalities of Linhares and Jaguaré (Espírito Santo, Southeastern Brazil; 19°06′ S, 39°45′ W and 19°18′ S, 40°19′ W; Figure 1). The RNV is adjacent to the Sooretama Biological Reserve (Reserva Biológica de Sooretama—RBS; 27,860 ha) and two other small Natural Heritage Private Reserves (Recanto das Antas, 2202 ha; and Mutum Preto, 379 ha). These reserves, together with other surrounding forest fragments, form a large remnant of native vegetation on flat terrain (Linhares-Sooretama Forest Block) of about 53,000 ha (about 11% of the forest remaining in the state [16]), which is intercepted in its central portion by the BR-101 Highway (Figure 1) and has documented direct impacts on fauna, including big cats [17].

Most of the reserve is covered by dense lowland rainforest located on flat terrain (‘Tabuleiro’ forest) and is classified as perennial seasonal forest [18]. In addition to dense lowland ‘Tabuleiro’ forests, a local forest vegetation type growing on sandy soils (called ‘Mussununga’), occasional native grasslands (called ‘Campo Nativo’), and vegetation associated with water bodies (riparian vegetation and marsh) are also present, composing a mosaic of habitats. The local climate is tropical with dry winters. The reserves are surrounded mostly by pastures and crops, especially fruit and coffee plantations, as well as forestry (*Eucalyptus*) plantations [18,19]. 

### 2.2. Trapping Procedures

We tried to trap jaguars and pumas (*Puma concolor*) between 26 September and 13 October 2018 (18 days) and between 19 June and 4 July 2019 (15 days). The methods authorized by the Brazilian environmental agencies for capturing large felids were foot-snares [6,12], the preferable method used in the country, and a minimum invasive capture system (MICS, described in detail in [10]; also see [13]). Despite foot-snares potentially posing a risk to animals and trappers, this capture technique is widely used and accepted for the capture of large felids (see review in [10]). The MICS has only previously been used to capture Eurasian lynx (*Lynx lynx* [13]), with no previous use on larger felids despite the potential advantages of the technique. Available data on camera-trapping records and indirect signals (feces and tracks) of mainly jaguars, and to a lesser extent pumas, obtained in the RNV during the two previous years to the trapping campaigns, were used to select sites for potential trapping.

Five foot-snare sets were installed at fixed sites following general procedures described in Araujo et al. [12]. The foot-snare sets were in a 5.2 km^2^ area, where most of the jaguar records were concentrated. Foot-snares were operated from sunset to sunrise and checked from a distance (between 800 and 2500 m) every hour using a Yagi antenna to check radio devices (VHF, Nortronic, model TAR-A5/B5) attached to foot-snares. One or more of these snare sets were moved if a large felid preyed upon a bait at other sites (see Section 3).

In addition to fixed sites with foot-snares, we set baits for large felids at different points spread over 50 km^2^ of the RNV (10 points in the first and 20 points in the second trapping campaign). Baits were live sheep weighing 15–25 kg in 2018 and 15–60 kg in 2019. Food and water were provided for the sheep every day in the morning (sometimes also in the afternoon if necessary). Baits were set mainly on the border of unpaved roads or jaguar paths in the shade. The sheep were tied to trees with 1.5–2.0 m of 8 mm thick propylene braided rope (in 2018) or double 10 mm thick round latex elastic cable (in 2019) attached to a collar of steel wire coated with rubber with twist-ties at both ends to protect the animals from unwanted entanglement. Some baits were relocated according to new evidence of a large felid presence detected during the campaign. At several sites with bait, we installed a camera trap (Bushnell, models Trophy Cam HD Aggressor Low-Glow e Trophy Cam HD Aggressor No-Glow) to identify eventual visitors and predators (species, size, and sex when possible). 

When a bait was preyed upon, we followed the protocol described in Palomares [10]. Briefly, when the bait was moved away by the felid (i.e., it managed to release the bait from the tree), we replaced the bait by relocating a sheep from another point (for safety reasons, we did not try to locate where the felid had moved the predated bait). If predated bait stayed on site, we protected it from vultures with branches until the afternoon. In both cases, all baits closer than 1.5 km to the predation site were removed, and one or two camera traps (Bushnell, same models) were installed if none were already at the site. One or more foot-snares and sometimes the MICS were installed on the same day the predated bait was discovered or the day after if it was not possible on the same day due to logistic or weather reasons.

### 2.3. Felid Immobilization and Collaring

Tiletamine hydrochloride plus zolazepam hydrochloride (Zoletil 100 ^®^) was used to immobilize felids using a dose of 7–8 mg/kg, and the estimated weight of the predator (initially 60 kg) was obtained from camera-trapping records. Once captured and immobilized, body weight and measurements were taken, and a Vectronic GPS radio-collar (https://www.vectronic-aerospace.com/accessed (accessed on 25 October 2017)) was installed and equipped with a drop-off system (see Section 3). The GPS collar was programmed to record one location every 4 h, and to perform an intensive 24 h tracking (with locations every 15 min) every week (from midday Saturday to midday Sunday) during a whole year, at which time the drop-off system should release the collar.

### 2.4. Data Analysis

We determined the types of vegetation or land use at recorded jaguar fixes based on the mapping available for the RNV and its surroundings [19], and the Google Maps platform for the points located inside the RBS and its surroundings. Home range size was calculated based on the 100% Minimum Convex Polygon (100%MCP) and the 95% adaptive Kernel density estimator (95%K) using the Standard Biweight (quartic) Kernel function. For MCP analysis, we first utilized all available fixes and, in a second approach, only the points located in the eastern portion of the BR-101 Highway were used. This was undertaken to correct distortions associated with MPC calculations by incorporating distant and infrequently utilized points (see Section 3). Home range analyses were performed using the program OpenJUMP (version 1.7.1 [20]).

## 3. Results

### 3.1. Trapping Campaign 2018

We set baits at 17 different points totaling 126 nights (baits active on average 7.4 nights per site, range = 2–15), and snares at seven different sites totaling 46 nights (active on average 6.6 nights per site, range = 3–10). At two of these points, foot-snares were set at bait sites (see below). Once, an adult male puma stopped, observed, and sniffed at a bait site, and on another occasion, an adult male jaguar passed closer (<3 m) to a bait. In the first case, a foot-snare was set for three consecutive nights, but no felids returned to the site. On another two occasions (one from an unknown large felid and another from the same adult male jaguar), baits were preyed upon. In both cases, felids moved the 15–20 kg preyed bait away from the site. On the first occasion, after bait replacement, we installed the MICS one night and one snare for four nights, but no felids returned to the site. In the five fixed foot-snare points installed in the 5.2 km^2^ area, the system was triggered twice (once by an unidentified large felid and another by a puma), while on two occasions a jaguar passed close by (<5 m) to the snare, and on another occasion a puma dodged the trap.

### 3.2. Trapping Campaign 2019

We only recorded the same adult male jaguar from the 2018 campaign. No records of pumas were obtained. We set baits at 24 different sites totaling 197 nights (on average, 8.2 nights per site, range = 1–14). On at least five occasions, the jaguar was <3 m from a bait. At two sites on three occasions, the jaguar predated the baits (two 50–60 kg and one 15–20 kg), and on the other two occasions, the jaguar was not apparently aware of the baits. 

We set foot-snares for 12 days at seven different sites, five at fix sites (a total of 14 nights, 2.8 nights per site on average, range = 2–4), and the other two at points where baits were preyed on by the male jaguar (see below; Figure 2). After predation on baits, foot-snare efforts were concentrated on these points. We installed foot-snares (1–3 sets) and the MICS at the same two sites where the male jaguar killed the baits on three different nights (see below; Figure 2). Furthermore, only foot-snares (1–3 sets) were installed at the same two points where the jaguar killed the baits on five different nights.

The jaguar passed <5 m or passed just over foot-snares without triggering the system in at least two of the fixed sites. In foot-snares set in baits preyed on by the jaguar, he did not return for five nights, and on three other nights, the jaguar triggered the system without being trapped (Figure 2). Regarding one of the nights when the jaguar did not return to trapped baits, he killed another bait at a site situated 3.4 km in a straight line from the first site and managed to remove the carcass. On the next night, the jaguar did not return, probably because he was feeding on the carcass remains in the place where he had dragged it. On two nights when the jaguar returned to the baits, he fled when he triggered foot-snares. On one of the nights, he did not return to the site during the rest of the night, and on the other night, he returned four hours later for slight feeding but did not trigger the snare again despite passing over it. According to camera trap records, he deflected his paw close to the snare. On the third occasion, he triggered one snare without capture when approaching the bait, but in this case, he stayed at the site, feeding on the bait after slightly moving it and inactivating the other two set foot-snares (Figure 2).

We installed the MICS on four occasions. On three occasions, it could not be used since the jaguar did not return to the baits or he fled after triggering a snare. On the fourth occasion, the MICS was used after the jaguar triggered the snares without trapping but stayed at the site feeding on bait (Figure 2; Appendix A). 

The night of capture (2 July at 23:55 h), we shot the MICS while staying at a distance of 300 m from the bait site, with a dart containing 500 mg Zoletil and impacting on the right hind quarter of the jaguar. When darted, the jaguar moved immediately from the site to the other side of the road (4 m from the carcass) and stayed there for 4–6 s looking at the carcass, until he decided to move into the forest (Appendix A). Fifteen minutes later, we located the jaguar lying down at 30 m from the bait site. We set an extra 500 mg Zoletil, weighed and measured him, and equipped him with a GPS-collar (Vertex Plus Iridium collar and 2D battery of Vectronic of 840 grs; https://www.vectronic-aerospace.com/vertex-plus-collar/ (accessed on 25 October 2017)). We left 2.5 h later when the jaguar showed a clear signal of recovery, although he did not leave the site until 5 h after the darting. The following dusk (at 18:34 h on 3 July), the jaguar returned to the carcass site and fed on the few remains left by vultures (Appendix A). The following nights, he did not return to the site, but stayed in the surrounding area (<2 km from capture point) for the following three days.

### 3.3. Jaguar Characteristics and GPS Tracking

The male jaguar weighed 80 kg, was in good condition and had no old or recent wounds. The body measurements were body length = 146 cm; tail length = 53 cm; height at withers = 67 cm; pelvic limb height = 64 cm; thoracic diameter = 96 cm; abdominal diameter = 119 cm; neck diameter (at base) = 61 cm; skull diameter = 61 cm; canine = 4.2 cm. He had intact canines and practically complete dentition (loss of only five incisors—1R 2R/1L 2L 3L). He was known in the area since 2005 through camera trap records when he was a subadult [21], and therefore his estimated age was 16 years old.

Drop-off worked as expected and the collar was released on 1 July 2020. The jaguar was located 4890 times, with 4798 3-Dimensional fixes (98.1%), therefore, location performance was high (86.3% of programmed fixes). After drop-off, the collar was still functioning normally until September 25 when we were able to go to the reserve to recover it after COVID-19 mobility restrictions had ended.

The jaguar maintained its activities mainly within protected areas (93.5% of fixes), mainly using native grassland, marsh, and dense lowland forest (76.8% of fixes; Table 1; Figure 1A). Similarly, most of the records obtained outside the reserves were in native vegetation (96.8%), with an emphasis on marsh areas (Table 1). Seven records occurred in firebreaks (Table 1), six on the border with forestry areas, and one within an agriculturally cultivated area. Only three records were obtained in areas intended for economic activities, but only at 28, 56, and 76 m from the forest edge (i.e., *Eucalyptus* plantations) (Table 1). On two occasions (February and April 2020), the jaguar crossed BR-101 and moved to the western end of the RBS. The home range, based on the MCP, was estimated as 658.7 km^2^ considering all fixes (Figure 1B) and 333.5 km^2^ considering only the points located in the eastern portion of the BR-101 Highway (Figure 1C). The home range size was 174.2 km^2^ using the 95%K (Figure 1D).

## 4. Discussion

The only jaguar detected during both trapping campaigns managed to avoid capture by foot-trap snares. However, the only and last chance to use the MICS successfully captured the jaguar. To our knowledge, the trapped male jaguar was naïve to any capture methods; despite this, he was aware of snares or quickly learned to avoid them. However, we cannot determine whether he had experience with other trap devices set by poachers. As predicted by Palomares [10], the MICS was relatively easy and quick to install, selective, and efficient. Additionally, an important factor was that the captured jaguar seemed stress-free as it never showed signs of awareness of human presence during the whole process or afterwards. Thus, the jaguar returned to the carcass to feed the following afternoon after capture as would be expected under normal circumstances [10,22], and he remained in the area for three additional days. Compared to other capture methods for large carnivores such as box-traps, leg-snare traps, foot-hold traps, or trained hounds, the use of the MICS should be promoted as it presents potential advantages mainly related to efficiency, selectivity, portability, reduced potential injury to animals or trappers, and animal stress (see [10,13] for more details about the MICS and advantages and constraints of the technique). However, one of the main points underpinning the successful use of the MICS is to facilitate a predictable return of the animals to the point of interest. In this regard, our experience with the capture reported here taught us that (1) baits must be well secured to prevent animals from moving them to another point where they will continue feeding during the following nights, thus impeding the use of the MICS, and (2) removing baits within 1.5 km of the predated one was not enough; in our case the male jaguar predated on another bait situated 3.4 km away and this impeded its return to the bait where snares were set.

The Vertex Plus Iridium GPS Collar’s performance was optimal (86.3% of expected programmed fixes were achieved with 3-Dimensional fixes), considering that jaguars moved very often within the forest, an environment where lower GPS location performance has been reported [23,24]. Additionally, battery duration and drop-off worked as expected. Therefore, these GPS radio-collars in Atlantic Forest habitats in Espírito Santo seemed to be optimal for GPS tracking of forest animals.

The home range of the male jaguar was within the observed range of values for the species, and similar to data for individuals living in similar environments and latitudes [25,26]. Nevertheless, the more intensively used area was only 33% of the protected area (based on the home range estimated with 95%K). Our records corroborate Morato et al. [27] by showing a jaguar preference for forests and areas close to watercourses in heavily forested landscapes and when waterways were close, respectively. As such, the trapped male jaguar defined its resource selection behavior as a function of the local forest cover and watercourses availability [27], resulting in most fixes within protected areas or in natural humid areas when outside reserves. Additionally, we suggest that the most used habitat types, such as dense lowland forest, native grassland, and marsh, should represent the areas with greater potential for locating prey by jaguars in the study area. In this regard, it is noteworthy that Tayassuidae and capybara (*Hydrochoerus hydrochaeris*) are among the main prey consumed by jaguars in the region [28]. Native grassland areas have been important for recording Tayassuidae by camera traps in the RNV (A.C. Srbek-Araujo, unpublished data), and capybaras are commonly found in areas associated with waterbodies, such as marshes.

Our study demonstrates that MICS should be considered a safe option for capturing Neotropical big cats, and corroborates with information that contributes to the understanding of natural jaguar history in the Atlantic Forest.

## Figures and Tables

**Figure 1 animals-13-03314-f001:**
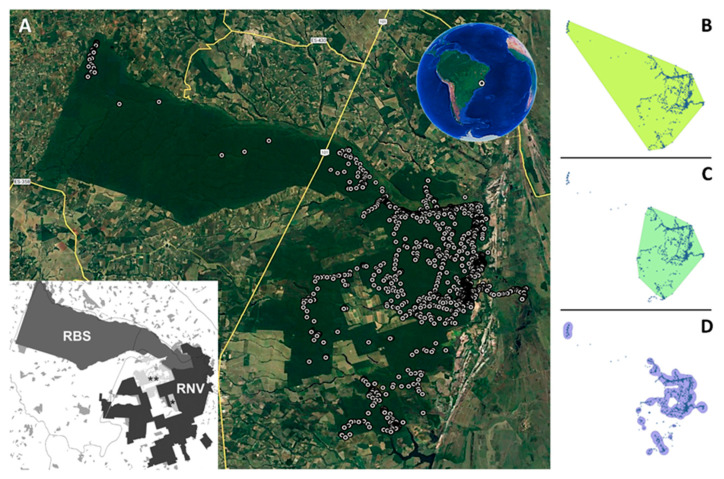
Map showing the location of the study area and the fixes of the jaguar captured in the Vale Natural Reserve (RNV) in July 2019, with the limits of Sooretama Biological Reserve (RBS), Mutum Preto Natural Heritage Private Reserve (*), and Recanto das Antas Natural Heritage Private Reserve (**) (Espírito Santo, Southeastern Brazil) (**A**). The MCP for the whole data set (**B**), only for the points located in the eastern portion of the BR-101 Highway (**C**), and the Kernel home range using 95% of fixes (**D**) are also shown. Highway BR-101 is the yellow line that crosses diagonally from the top to the bottom (through the middle) of the main map. The satellite image was obtained from Google Earth (Image © 2023 Airbus, Image © 2023 CNES/Airbus, Image © 2023 Maxar Technologies).

**Figure 2 animals-13-03314-f002:**
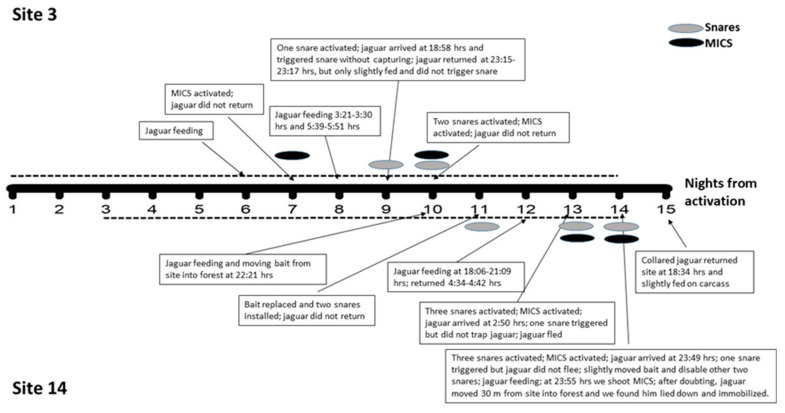
Sequence of events and actions at two sites where the adult male jaguar preyed on baits (sites 3 and 14) in the Vale Natural Reserve (Espírito Santo, Southeastern Brazil) between 19 and 20 June (night 1) and 3 and 4 July (night 15) 2019. The broken line represents days when baits were available at both sites.

**Table 1 animals-13-03314-t001:** Vegetation or land use types at locations of the male jaguar as tracked by GPS-collar in the Linhares-Sooretama Forest Block, Southeastern Brazil (3 July 2019 to 1 July 2020).

Vegetation or Land Use Type	Protected Reserves	%	Outside Reserves	%	Total	%
Native grassland	1295	28.9	10	3.2	1305	27.2
Mussununga forest	245	5.5	79	25.5	324	6.8
Dense lowland forest	1157	25.8	29	9.4	1186	24.7
Riparian vegetation	731	16.3	50	16.1	781	16.3
Marsh	1060	23.6	132	42.6	1192	24.8
Firebreak	-	-	7	2.3	7	0.2
Forestry	-	-	3	0.9	3	0.1
Total	4488	93.5	310	6.5	4798	-

## Data Availability

There is no data to share.

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
