# Peer review of "First Capture of a Jaguar Using a Minimally Invasive Capture System for GPS Tracking in an Isolated Patch of Atlantic Forest in Southern Brazil"

_animals, 2023, doi:10.3390/ani13213314_

Round 1
Reviewer 1 Report
This is an interesting and useful paper on a highly relevant topic. The work is nicely connected to the wider international literature.
The MICS method worked well and the data obtained on home range (especially in relation to the highway) is valuable.
I was a bit concerned by the use of live baits but I noted at the end that it went through the relevant Ethics and Animal Welfare Committee at the University of Vila Velha. Such live baits could probably not be used in some countries and I wonder if this is a concern for more widespread use of MICS?
I enjoyed the video very much and watched it multiple times.

I made the comment that the English needs improving. Actually overall the English quality is very good. However, there were some very minor changes needed. I have done a full edit of the manuscript - see attached. If you are happy with the changes I suggest then I think the English will be fine. I am a native English speaker (but not perfect!).
Author Response
Thank you very much for the review.
All suggestions and comments have been considered and included in the revised version. It is easy to track in the revised version.
Author Response
In the attached document I details all the changes performed in relation with the suggestion of the reviewer 2. Nevertheless, I advance that I followed all suggestions with the exception of including a figure with the location of trapping or potentially trapping point. Some clarifications have also added.

Reviewer 3 Report
Please see enclosed file.

Please see details in enclosed file to authors
Author Response
First capture of a jaguar using a minimally invasive capture system for GPS tracking in an isolated patch of Atlantic Forest in southern Brazil

Round 2
Reviewer 3 Report
Please, see enclosed file

Still a minor change required. Please, see enclosed file.
Author Response
All suggestions have been considered:
Line 330: Please change the time of the verb to past (“gave us”, or “provided us”)).
THIS WAS ALREADY CORRECTED IN THE ENGLISH VERSION
Lines 336 onward: The format of the references should be revised to fit the
requirements of the journal (ej. names of the journals in italics, volume number in bold,
etc)
REVISED!
Regarding Editor specific query:
Ref #1 (line 39): It seems adequate, but not necessary. In our opinion, a much better reference would be the following review/book: Large carnivores and the conservation of the biodiversity, by Ray, J.C. et al. Island Press, 2005.
REFERENCE 1 HAS BEEN REPLACED BY THIS OF RAY ET AL.
Ref #10 (lines 43 and 46 ): In our opinion might be not adequate in this first sentence (line 43) but necessary for sentence starting line 46. Please note that this reference appears cited at least 8 times in the ms, and up to 3 times in a single paragraph, lines 101, 103, 105. That´s seems quite repetitive and unnecessary. I suggest to revise the ms and remove those unnecessary.
REFERENCE 10 IS CRUCIAL TO UNDERSTAND THE CONTENT OF THE PAPER. NEVERTHELESS, IT HAS BEEN REMOVED FROM LINES 56, 104, AND 271.